DOI: 10.1038/s41467-018-05966-z · OPEN

# SIRT6 haploinsufficiency induces BRAF$^{V600E}$ melanoma cell resistance to MAPK inhibitors via IGF signalling

Thomas Strub[1,2], Flavia G. Ghiraldini[1,2], Saul Carcamo[1,2,3], Man Li[4], Aleksandra Wroblewska[5], Rajendra Singh[2,6], Matthew S. Goldberg[2,6], Dan Hasson[1,2], Zichen Wang [7], Stuart J. Gallagher[8,9], Peter Hersey[8,9], Avi Ma'ayan [7], Georgina V. Long[9,10,11], Richard A. Scolyer [9,10,12], Brian Brown[3,5,13], Bin Zheng[8] & Emily Bernstein[1,2,3]

While multiple mechanisms of BRAF$^{V600}$-mutant melanoma resistance to targeted MAPK signaling inhibitors (MAPKi) have been reported, the epigenetic regulation of this process remains undetermined. Here, using a CRISPR–Cas9 screen targeting chromatin regulators, we discover that haploinsufficiency of the histone deacetylase SIRT6 allows melanoma cell persistence in the presence of MAPKi. Haploinsufficiency, but not complete loss of SIRT6 promotes IGFBP2 expression via increased chromatin accessibility, H3K56 acetylation at the *IGFBP2* locus, and consequent activation of the IGF-1 receptor (IGF-1R) and downstream AKT signaling. Combining a clinically applicable IGF-1Ri with BRAFi overcomes resistance of SIRT6 haploinsufficient melanoma cells in vitro and in vivo. Using matched melanoma samples derived from patients receiving dabrafenib + trametinib, we identify IGFBP2 as a potential biomarker for MAPKi resistance. Our study has not only identified an epigenetic mechanism of drug resistance, but also provides insights into a combinatorial therapy that may overcome resistance to standard-of-care therapy for BRAF$^{V600}$-mutant melanoma patients.

[1] Department of Oncological Sciences, Icahn School of Medicine at Mount Sinai, One Gustave L. Levy Place, New York, NY 10029, USA. [2] Department of Dermatology, Icahn School of Medicine at Mount Sinai, One Gustave L. Levy Place, New York, NY 10029, USA. [3] Graduate School of Biomedical Sciences, Icahn School of Medicine at Mount Sinai, One Gustave L. Levy Place, New York, NY 10029, USA. [4] Cutaneous Biology Research Center, Massachusetts General Hospital, Harvard Medical School, Charlestown, MA 02129, USA. [5] Department of Genetics and Genomic Sciences, Mount Sinai Center for Bioinformatics, Icahn School of Medicine at Mount Sinai, One Gustave L. Levy Place, New York, NY 10029, USA. [6] Department of Pathology, Mount Sinai Center for Bioinformatics, Icahn School of Medicine at Mount Sinai, One Gustave L. Levy Place, New York, NY 10029, USA. [7] Department of Pharmacological Sciences, Mount Sinai Center for Bioinformatics, Icahn School of Medicine at Mount Sinai, One Gustave L. Levy Place, New York, NY 10029, USA. [8] Centenary Institute, Camperdown NSW 2050, The University of Sydney, Sydney, Australia. [9] Melanoma Institute Australia, Wollstonecraft NSW 2065, The University of Sydney, Sydney, Australia. [10] Sydney Medical School, University of Sydney, Sydney, NSW 2050, Australia. [11] Royal North Shore Hospital, Sydney, NSW 2065, Australia. [12] Royal Prince Alfred Hospital, Sydney, NSW 2050, Australia. [13] Precision Immunology Institute, Icahn School of Medicine at Mount Sinai, One Gustave L. Levy Place, New York, NY 10029, USA. These authors contributed equally: Flavia G. Ghiraldini, Saul Carcamo. Correspondence and requests for materials should be addressed to E.B. (email: emily.bernstein@mssm.edu)

The incidence of cutaneous malignant melanoma is rising and its therapeutic management remains challenging[1]. In recent years, there has been extensive therapeutic development to inhibit key biological targets, such as constitutively activated BRAF (BRAF$^{V600E/K}$) and its downstream effectors MEK and ERK[2–4]. Although a large proportion of patients with advanced metastatic melanoma harboring BRAF$^{V600E/K}$ mutation respond to MAPKi, subsequent resistance remains a major clinical challenge[5]. While a variety of genetic mutations, amplifications, and splicing alterations have been described in acquired resistance to MAPKi[6], these mechanisms account for only a fraction of cases. Notably, the epigenetic mechanisms of melanoma drug resistance remain poorly understood.

Emerging evidence suggests that chromatin-mediated processes are linked to the development and progression of cancer. Our group and others have revealed a key role for histone variants[7,8], histone deacetylases[9–12], histone methyltransferases[13–16], histone readers[17,18], chromatin remodeling complexes[19,20], or DNA hydroxymethylation (5-hmC)[21] in the pathogenesis of melanoma. Further, a growing body of evidence suggests that altered chromatin states can modulate the response to targeted therapies in multiple tumor types[22,23]. Relevant to our study, recent reports have implicated DNA methylation, transcriptional changes, microRNA alterations, as well as microenvironmental stressors in promoting melanoma drug resistance to MAPKi in BRAF$^{V600}$-mutant melanoma[24–30], suggesting non-genetic mechanisms of plasticity of melanoma tumors to overcome these therapies. Moreover, it suggests that epigenetic alterations may play a key role in rewiring the chromatin landscape of melanoma cells to allow adaptation to MAPKi. Thus, shedding light onto the transcriptomic and epigenetic alterations underlying acquired MAPKi resistance in melanoma is of critical importance.

In order to probe the chromatin-mediated mechanisms involved in melanoma resistance to MAPKi, here we perform a CRISPR–Cas9 screen in BRAF$^{V600E}$ human melanoma cells targeting chromatin modifiers in the context of MAPKi. We identify SIRT6 as a regulator of resistance to the clinically relevant BRAF inhibitor (BRAFi), dabrafenib, or combination dabrafenib + trametinib (MEK inhibitor, MEKi) in BRAF$^{V600E}$ melanoma. Through integrated transcriptomic, proteomic, and epigenomic analyses, we discover that SIRT6 haploinsufficiency increases IGFBP2 expression and promotes melanoma cell survival through the activation of IGF-1R/AKT signaling. In contrast, complete loss of SIRT6 does not promote IGFBP2 expression, but rather allows sensitivity to MAPKi through a DNA damage response. Collectively, our study provides information on: (1) a previously unknown epigenetic mechanism of melanoma drug resistance, (2) a dose-dependent effect of SIRT6 levels on the drug resistance phenotype, and (3) a combinatorial therapy that may overcome resistance to MAPKi for a subset of BRAF$^{V600}$-mutant melanoma patients.

## Results

### A CRISPR–Cas9 screen identifies histone acetylation modifiers in melanoma MAPKi resistance.

We performed a CRISPR–Cas9 screen targeting ~140 chromatin factors containing enzymatic activity in BRAF$^{V600E}$ human melanoma cells (Fig. 1a, Supplementary Fig. 1a, Supplementary Data 1). SKMel-239 cells stably expressing Cas9 were infected with the single-guide RNA (sgRNA) library (3–4 sgRNAs per gene encoded in pLKO.1-EGFP); GFP-positive cells were sorted for expansion (Fig. 1a) and cultured with DMSO (control), dabrafenib, or dabrafenib + trametinib for 6 weeks (Fig. 1a). While the majority of cells were sensitive to MAPKi[31], a fraction of cells survived the drug treatments. Genomic DNA was isolated from all conditions, including control cells at days 0 and 42, and the abundance of each sgRNA was determined using next-generation sequencing (Fig. 1a, Supplementary Fig. 1b). As expected from the strong selection of the screen, the sgRNA distribution of drug-treated cells at 6 weeks was significantly different than control cells (Supplementary Fig. 1b).

Our screen revealed genes whose depletion conferred resistance to MAPKi, with enrichment of enzymes that mediate histone acetylation. These include the histone acetyltransferases (HATs), KAT1 (HAT1), and KAT2B (PCAF), as well as the NAD+-dependent histone deacetylase SIRTUIN 6 (SIRT6) (Fig. 1b). We focused on SIRT6, which has been reported to act as a tumor suppressor[32], but has not been implicated in melanoma resistance. Functionally, SIRT6 deacetylates a variety of protein substrates, including histone H3 at lysines 9 and 56 (H3K9ac and H3K56ac), and regulates cellular metabolism as well as DNA damage responses[32–36]. We validated the functional impact of SIRT6 deficiency on drug resistance by introducing individual sgRNAs, and deriving clonal cell lines (Fig. 1c, d, Supplementary Fig. 1c). First, using the precise sgRNA identified in both screens (i.e., SIRT6.2, Fig. 1b), we found that SIRT6 protein levels were reduced, but still detectable (Fig. 1c). Consistent with this finding, we found mono-allelic editing of the SIRT6 genomic locus (Supplementary Fig. 1d-f). To further confirm a role for SIRT6 in melanoma drug resistance, we generated SKMel-239 MAPKi-resistant cells by prolonged drug treatment[37], and observed decreased SIRT6 levels in all resistant clones (Fig. 1e), indicating that lower levels of SIRT6 are associated with acquired resistance. By examining short-term cultures (STCs) derived from paired BRAF$^{V600E}$ tumor biopsies collected prior to treatment (Pre) and upon onset of resistance to vemurafenib (Prog) from three melanoma patients[38], we observed decreased levels of SIRT6 in two out of three samples upon resistance (Fig. 1f). Similar results were observed for KAT2B and KAT1 (Supplementary Fig. 1g), consistent with the hits identified in our screen.

### SIRT6 haploinsufficiency in BRAF$^{V600E}$ melanoma cells decreases sensitivity to MAPKi.

To further investigate SIRT6, we generated additional CRISPR–Cas9 edited clonal cell lines that exhibited either decreased or complete loss of SIRT6 (Fig. 2a, top panel). As expected from our validation studies (Fig. 1c), clones with decreased levels of SIRT6 displayed MAPKi resistance (Fig. 2a, bottom panel). Unexpectedly, however, clones devoid of SIRT6 were as sensitive, if not more sensitive, to MAPKi as control cells (Fig. 2a). Strikingly, similar results were observed in additional BRAF$^{V600E}$ melanoma cell lines (Supplementary Fig. 2a, b). Next, we aimed to reproduce the reduction, but not complete loss of SIRT6 levels using shRNAs. Knockdown of SIRT6 in several BRAF$^{V600E}$ melanoma cell lines and an STC exhibited resistance to MAPKi (Fig. 2b, Supplementary Fig. 2c, d). Notably, the clonal cell lines SIRT6.2–7, SIRT6.1–1, as well as SIRT6 knockdown cells, displayed similar growth rates as their respective controls (Supplementary Fig. 2e), suggesting that their differential sensitivity to the inhibitors is independent of proliferation.

To investigate whether decreased expression of SIRT6 mediates the response to BRAFi in vivo, two resistant clones (SIRT6.2–7 and SIRT6.4–1) were injected into nude mice and grown as xenografts to assess their response to vemurafenib[29]. As expected, while control tumors were sensitive to the drug, SIRT6 haploinsufficient tumors were significantly less so (Fig. 2c). Collectively, these in vitro and in vivo results suggest that partial suppression of SIRT6 confers BRAF$^{V600E}$ melanoma cells the ability to persist in the presence of MAPKi.

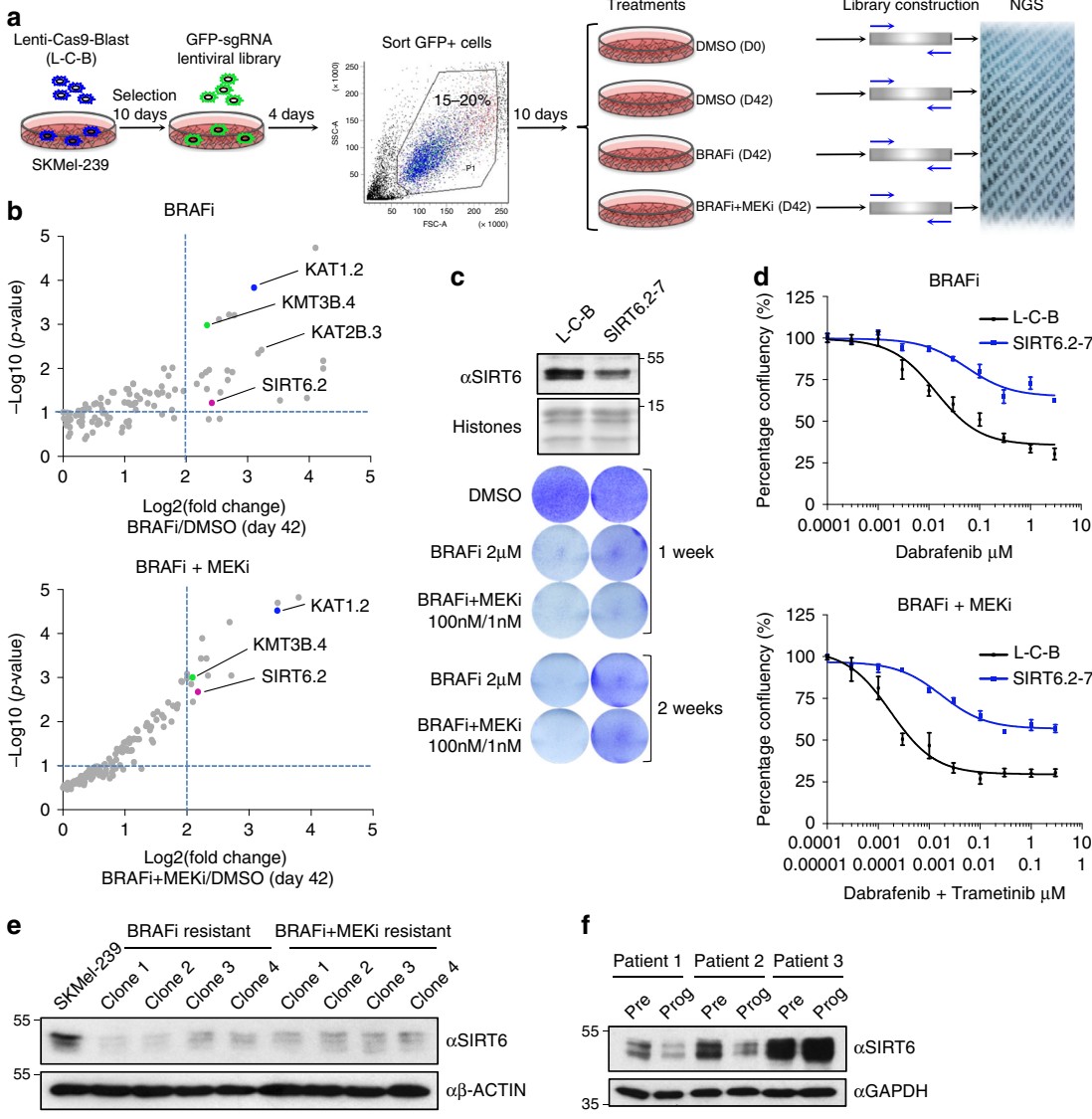

**Fig. 1** CRISPR–Cas9 screen identifies SIRT6 as a determinant of melanoma drug resistance. **a** Schematic of the CRISPR–Cas9 screen for chromatin factors that regulate dabrafenib (BRAFi) and dabrafenib + trametinib (BRAFi + MEKi) resistance in SKMel-239 BRAF[V600E] melanoma cells. **b** Scatterplot of enrichment of sgRNAs after 6 weeks of BRAFi (top) or BRAFi + MEKi treatment (bottom). Genes in the upper right quadrant represent significant hits in each screen and those indicated in color represent significant hits present in both. **c** (Top) Immunoblot of SIRT6 in the indicated whole-cell lysate samples in SKMel-239. Histones used as a loading control. L-C-B and SIRT6.2–7 cells were seeded at the same density and cultured in DMSO or in the presence of the MAPKi as indicated for 1 or 2 weeks. **d** Growth inhibition curves are shown for BRAFi (top), as well as BRAFi + MEKi (bottom) at 72 h of treatment (n = 3). Data are mean ± SEM. **e**. Immunoblot of SIRT6 in the indicated whole-cell lysates of SKMel-239 or MAPKi-resistant clones generated after continuous exposure to the indicated drugs. B-Actin was used as a loading control. **f** Immunoblot of SIRT6 in the indicated whole-cell lysates of patient-derived short-term cultures (STCs; matched samples before vemurafenib treatment (Pre) and upon disease progression (Prog)). GAPDH was used as loading control

**SIRT6 haploinsufficiency promotes MAPKi resistance in an ERK-independent manner**. Because SIRT6 is a chromatin-associated histone deacetylase[35,39], we next investigated whether reduced levels of SIRT6 altered H3 acetylation. Under baseline conditions (i.e., no inhibitors), we observed a dose-dependent increase of only H3K56ac in both mono- and bi-allelically edited SIRT6 clones (Fig. 2d, compare lanes 1, 4, 7) as reported[39]. However, H3K56ac further increased in the presence of drugs only in the SIRT6.1-1 clone (Fig. 2d, lanes 8, 9). These high levels of H3K56ac are consistent with increased DNA damage detected by comet assay in SIRT6 null cells (Supplementary Fig. 3c) and previous reports[34,36]. Overall,

these data suggest that H3K56ac is the primary target of SIRT6 in the context of melanoma.

Since ERK signaling reactivation is the primary mechanism of resistance in response to MAPKi[40], we next sought to determine whether SIRT6 reduction promotes ERK signaling, and is thus responsible for the resistance observed in SIRT6.2–7 cells. Intriguingly, decreased SIRT6 expression did not alter the levels of MEK or ERK phosphorylation compared to control cells in either short (e.g., 6 h) or extended (4 days) time frames (Fig. 2e, f). Together, these data suggest that SIRT6 haploinsufficiency in BRAF[V600E] melanoma cells decreases sensitivity to MAPKi independent of ERK signaling. Of note, we also validated the

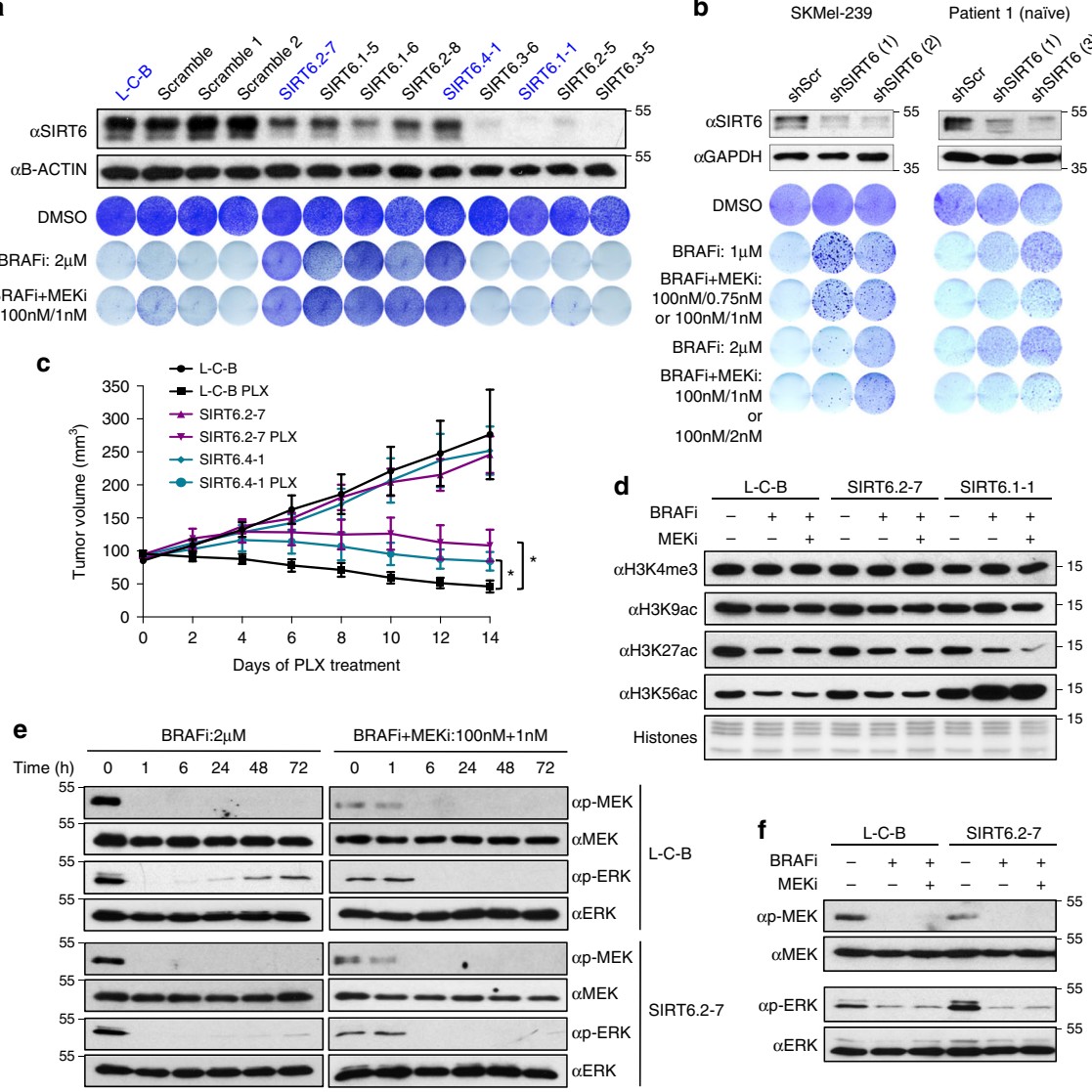

**Fig. 2** SIRT6 haploinsufficiency promotes resistance to MAPKi in BRAF^V600E melanoma cells. **a** Immunoblot of SIRT6 in the indicated whole-cell lysates of SKMel-239 clonal CRISPR cell lines. B-Actin was used as a loading control (top; CRISPR clones used for follow-up experiments are indicated in blue). SKMel-239 cells were seeded at the same density and cultured in DMSO (1 week) or in the presence of the MAPKi (2 weeks) as indicated (bottom). **b** Immunoblot of SIRT6 in the indicated whole-cell lysates in SKMel-239 (left) and Patient 1 treatment-naive STC (right). GAPDH was used as a loading control. SKMel-239 and Patient 1 cells infected with control shRNA (scrambled) or shRNAs against SIRT6 (1, 2, or 3) were seeded at the same density and cultured in DMSO (1 week) or in the presence of MAPKi (4 and 5 weeks, respectively). **c** Quantification of tumor volume in nude mice bearing xenograft tumors of L-C-B, SIRT6.2–7, or SIRT6.4–1 cells that were fed a control or vemurafenib (PLX)-containing diet. Mann–Whitney test was performed for comparison between the three groups of mice treated with PLX. Data are mean ± SEM. *$P < 0.05$. $n = 6$-7. **d** The effect of BRAFi or BRAFi + MEKi treatment on H3K4me3, H3K9ac, H3K27ac, and H3K56ac in the indicated CRISPR cell lines after 4 days of treatment ± 2 µM of BRAFi or 100 nM BRAFi + 1 nM MEKi. Histones used as a loading control. **e** The effects of BRAFi or BRAFi + MEKi treatment for the indicated periods of time on components of the MAPK signaling pathway in the indicated CRISPR cell lines. **f** Immunoblot of 2 µM of BRAFi or 100 nM BRAFi + 1 nM MEKi treatment for 4 days on components of the MAPK signaling pathway in L-C-B and SIRT6.2–7 cells

functional impact of KAT2B deficiency on drug resistance where similar results were observed, supporting alternative resistance mechanisms through these chromatin modifiers (Supplementary Fig. 4a, b).

**IGFBP2 is a direct SIRT6 target that is upregulated upon haploinsufficiency**. To understand how SIRT6 participates in drug resistance, we performed transcriptomic (RNA-seq) analyses of L-C-B (control), SIRT6.2–7 and SIRT6.1-1 cells in the absence or presence of MAPKi. First, our transcriptomic analyses of control cells identified 2178 MAPKi-sensitive genes (i.e.,

downregulated) with enrichment of cell cycle genes (Fig. 3a, Supplementary Fig. 5a, b), in accordance with the proliferation arrest and cell death observed upon MAPKi as previously reported[31,41]. We next compared these MAPKi-sensitive genes with 864 and 222 upregulated genes identified in SIRT6.2–7 cells under control or MAPKi conditions, respectively (Fig. 3a). We identified molecular functions such as transmembrane RTK (receptor tyrosine kinase) and growth factor binding (Supplementary Fig. 5c-e). This integrated strategy aimed to identify upregulated SIRT6 targets/pathways whose inhibition might prevent resistance to MAPKi. We identified 20 genes (e.g., *IGFBP2*, *AEBP1*, *SPOCK1*, etc.) that potentially promote

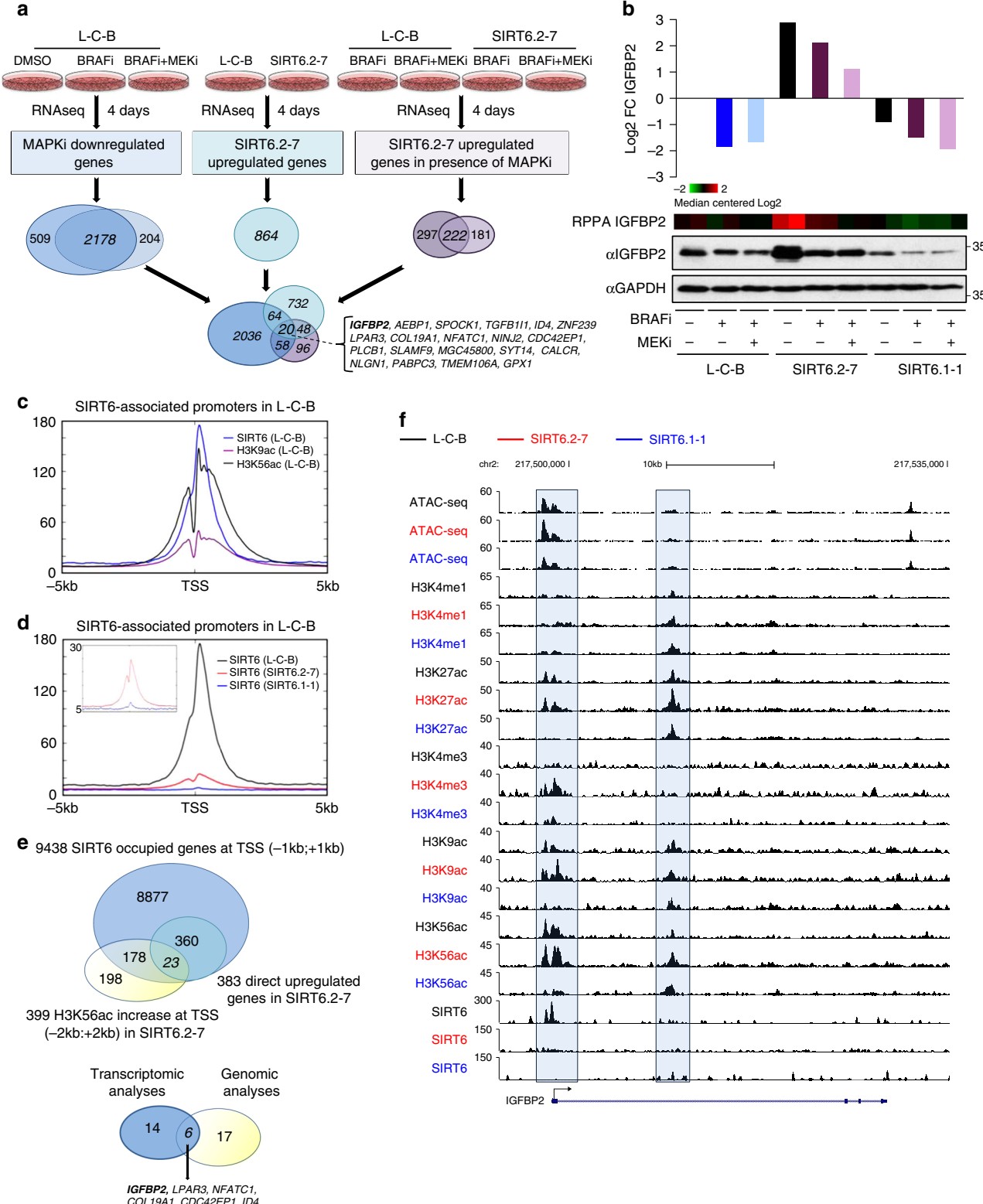

resistance to MAPKi upon SIRT6 haploinsufficiency (Fig. 3a, Supplementary Data 2), some of which were validated by quantitative RT-PCR (qPCR) (Supplementary Fig. 5f). Importantly, some of these genes have been implicated in resistance to BRAFi in melanoma[28,42], reinforcing our strategy.

Strikingly, however, the majority of these genes are downregulated in SIRT6.1-1 cells (Supplementary Data 2). Consistent with the comet assay data (Supplementary Fig. 3c), gene ontology analyses for SIRT6.1-1 upregulated genes (vs. control) in the absence of MAPKi showed enrichment for a DNA damage checkpoint, while downregulated pathways included key functions such as transcriptional regulation and axon guidance (Supplementary Fig. 6a, b, Supplementary Data 2). Moreover, SIRT6.1-1 cells essentially lost the ability to grow as xenografted

**Fig. 3** IGFBP2 is a SIRT6 target that is upregulated in MAPKi melanoma resistance. **a** Schematic of the integrated transcriptional profiling of MAPKi (left) on SKMel-239 L-C-B or SIRT6.2–7 and the SIRT6 regulatory network in the absence (middle) or presence (right) of MAPKi. 2178 shared downregulated genes were identified as MAPKi-sensitive genes; 864 upregulated genes were identified in SIRT6.2–7 cells; 222 shared upregulated genes upon MAPKi treatment in SIRT6.2–7 cells. Twenty genes identified as potential targets are shown in the bracket. **b** Haploinsufficiency of SIRT6 increases IGFBP2 expression. $Log_2$ fold change from RNA-seq for IGFBP2 (top); blue: L-C-B with the indicated drugs or DMSO; black: DMSO-treated cells; purple: SIRT6.2–7 or SIRT6.1-1 with the indicated drugs. IGFBP2 protein expression after 4 days of 2 μM of BRAFi or 100 nM BRAFi + 1 nM MEKi in the indicated CRISPR cell lines by RPPA analysis (middle), represented by heatmap (Normalized Log2 Median Centered is shown) and by immunoblot (bottom). GAPDH was used as a loading control. **c** SIRT6, H3K9ac, and H3K56ac ChIP-seq meta-profiles in L-C-B cells at SIRT6-associated promoters. Plot represents average read counts per 10 bp bins. **d** ChIP-seq meta-profiles for SIRT6 in L-C-B, SIRT6.2–7, and SIRT6.1-1 cell lines at SIRT6-associated promoters. Upper left quadrant shows zoom in for SIRT6 signal in SIRT6.2–7 and SIRT6.1-1 at SIRT6-bound promoters. Plot represents average read counts per 10 bp bins. **e** Venn diagram displaying SIRT6-occupied genes at TSSs (transcription start sites) associated with increased H3K56ac in SIRT6.2–7 cells. 399 TSS show an increase in H3K56ac in SIRT6.2–7 cells and 201 SIRT6-bound TSS show an increase at H3K56ac in SIRT6.2–7 cells compared to L-C-B. 23 SIRT6-bound TSS show an increase for H3K56ac and are upregulated in SIRT6.2–7 vs. L-C-B (top). Venn diagram displaying SIRT6 targets identified by integrating transcriptomic and genomic analyses (bottom). **f** Capture of the UCSC (GRCh37/hg19) genome browser showing the *IGFBP2* locus. L-C-B (black), SIRT6.2–7 (red), and SIRT6.1-1 (blue) ATAC-seq and ChIP-seq profiles for H3K4me1, H3K27ac, H3K4me3, H3K9ac, H3K56ac, and SIRT6 are shown (Reads Per Kilobase per Million reads). Genomic coordinates shown on top and *IGFBP2* locus on bottom

tumors, suggesting survival deficiencies in vivo (Supplementary Fig. 6c). These collective data are consistent with the observed sensitivity of SIRT6.1-1 cells to MAPKi.

We also performed Reverse Phase Protein Array (RPPA) of L-C-B (control), SIRT6.2–7, and SIRT6.1-1 cells in the absence or presence of MAPKi (Supplementary Data 3). This RPPA is an antibody-based array that includes ~300 antibodies to detect total protein levels or specific post-translational modifications of proteins[43]. Notably, this study revealed one upregulated candidate in common with the transcriptomic analyses in SIRT6.2–7 cells, namely IGFBP2. This was confirmed by immunoblot; IGFBP2 protein levels were elevated in SIRT6.2–7 cells in both untreated and drug-treated cells, consistent with their ability to persist in the presence of MAPKi, while IGFBP2 levels were lower than control cells for SIRT6.1-1 (Fig. 3b), consistent with their sensitivity to MAPKi.

As SIRT6 is a chromatin-associated deacetylase, we hypothesized that it could directly influence gene expression of its target genes by modifying chromatin structure. We performed ATAC-seq, ChIP-seq for SIRT6, as well as histone modifications in L-C-B, SIRT6.2–7, and SIRT6.1-1 cells, including H3K4me1, H3K4me3, H3K9ac, H3K27ac, as well as the main target site of SIRT6 deacetylation, H3K56ac. We discovered that among SIRT6-occupied loci, 56% were at promoters, among which 75% co-localize with H3K9ac and H3K56ac; 27% within gene bodies; and 17% in distal regions (Fig. 3c, Supplementary Fig. 7a). Only 30% of these latter two categories co-localize with H3K9ac and H3K56ac (Supplementary Fig. 7a). Thus, the majority of SIRT6-binding sites were at promoters, enrichment for which was reduced in the SIRT6.2–7 cells and nonexistent in SIRT6.1-1 cells (Fig. 3d).

To identify direct SIRT6 targets/pathways whose inhibition might prevent resistance to MAPKi, we integrated SIRT6-bound promoters identified by ChIP-seq in control cells with the transcriptome of SIRT6.2–7 cells and identified 325 downregulated and 383 upregulated genes bound by SIRT6 around the transcription start site (TSS) (Supplementary Fig. 7b). We focused on the 383 SIRT6-occupied upregulated genes, which showed enrichment for specific molecular functions such as receptor and growth factor binding (Supplementary Fig. 7c). As SIRT6 deacetylates H3K56ac, we next sought to identify those genes bound by SIRT6 in control cells that display increased H3K56ac at their TSS in SIRT6.2–7 cells (Fig. 3e). Although we did not observe a global increase in H3K56ac at the promoters of such genes in SIRT6.2–7 cells (Supplementary Fig. 7d), we identified 201 genes bound by SIRT6 with increased H3K56ac, among which, 23 were upregulated in SIRT6.2–7 cells (Fig. 3e). Finally,

integration of genes that were SIRT6-bound, upregulated in SIRT6.2–7 cells, and displayed increased promoter H3K56ac (Supplementary Data 4) with the 20 genes identified to facilitate resistance to MAPKi (Fig. 3a), led us to six candidates of which we focused on *IGFBP2* (Fig. 3e, f, Supplementary Data 4). We also noted increased H3K4me3 signal and open chromatin at the *IGFBP2* TSS, consistent with increased mRNA expression observed in SIRT6.2–7 cells (Fig. 3b, f). A putative enhancer marked by H3K4me1 and H3K27ac within the *IGFBP2* locus also showed increased H3K56ac (Fig. 3f). Importantly, chromatin accessibility and marks of active transcription were reduced at the *IGFBP2* TSS in SIRT6.1-1 cells, consistent with their sensitivity to MAPKi and supporting a role for IGFBP2 expression in promoting resistance to MAPKi (Fig. 3f). Taken together, our data suggest that SIRT6 directly regulates the *IGFBP2* locus and the levels of SIRT6-binding appear to determine the transcriptional output of this locus.

**Co-targeting of MAPK and IGF-1R signaling impedes melanoma drug resistance.** Because IGFBP2 controls the bioavailability of IGFs, which in turn modulates IGF-1R/IR (insulin receptor) signaling pathways[44,45], we next queried whether IGFBP2 plays a role in the MAPKi resistance phenotype. To this end, we treated control cells with insulin and/or ectopic expression of IGFBP2, which revealed decreased sensitivity to MAPKi when both were applied (Fig. 4a, Supplementary Fig. 8a). Consistent with this, IGFBP2 protein levels correlate with resistance to MAPKi in several BRAF^V600^-mutant melanoma cell lines (Supplementary Fig. 8b). Next, we tested whether IGF-1R and its downstream effectors (ERK and AKT) were activated in SIRT6.2–7 cells in the absence or presence of BRAFi. No changes were observed under baseline conditions (Fig. 4b). As mentioned above (Fig. 2e, f), MEK and ERK phosphorylation were similar between control and SIRT6.2–7 cells (Fig. 4b). However, upon BRAFi, SIRT6.2–7 cells showed elevated levels of phosphorylated IGF-1R and AKT compared to control L-C-B cells (Fig. 4b). Therefore, in the context of SIRT6 haploinsufficiency, the IGF-1R/AKT survival pathway is activated, consistent with a MAPK-independent resistance mechanism.

We next tested whether inhibiting IGF-1R signaling might suppress MAPKi resistance using linsitinib, which inhibits ligand-stimulated autophosphorylation of IGF-1R and IR[46]. Combination treatment of SIRT6.2–7 cells with dabrafenib + linsitinib led to a substantial decrease of IGF-1R and AKT phosphorylation (Fig. 4b). Importantly, this combination significantly reduced proliferation and led to increased apoptosis compared to cells treated with dabrafenib or linsitinib alone (Fig. 4c, d,

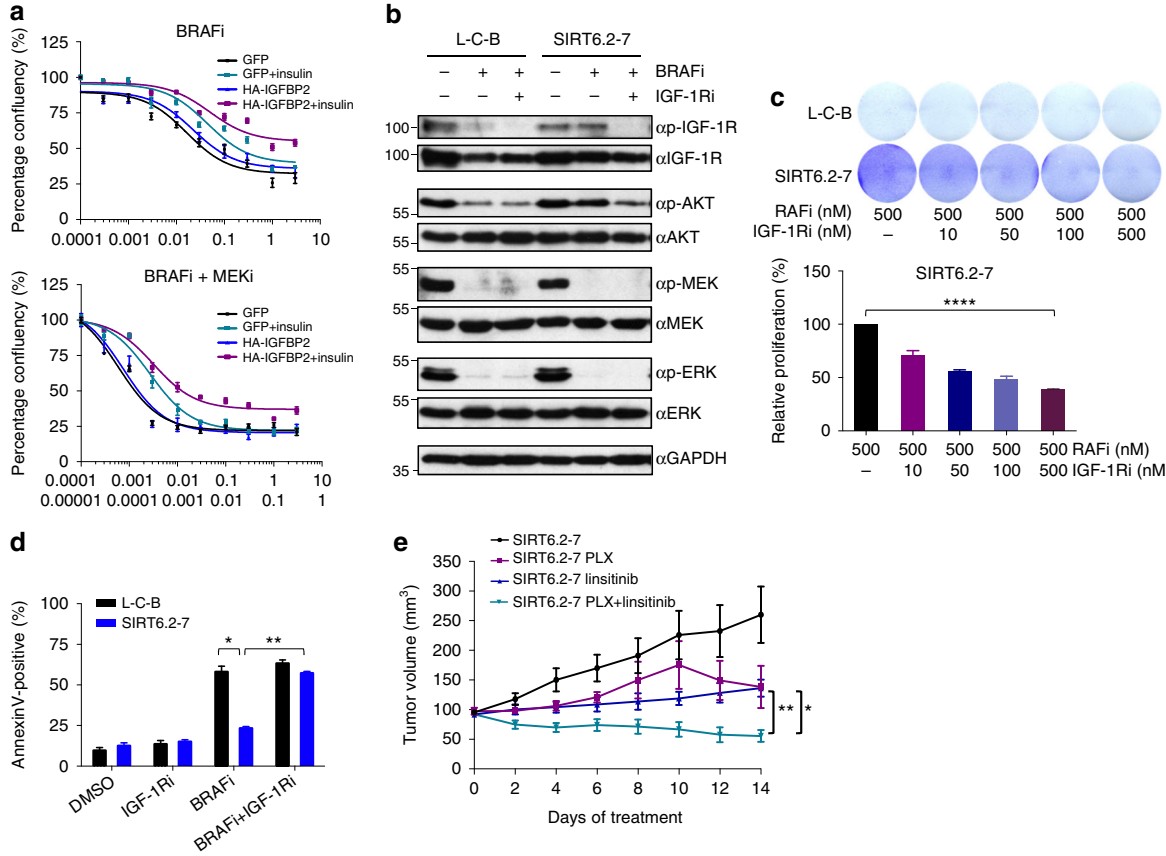

**Fig. 4** Inhibition of both MAPK and IGF-1R signaling impairs drug resistance. **a** Growth inhibition curves for BRAFi (top) and BRAFi + MEKi (bottom) in the indicated conditions (μM) after 72 h, $n = 3$. Data are mean ± SEM. **b** The effects of BRAFi or BRAFi + IGF-1R/IRi treatment on components of IGF-1R, AKT, and ERK signaling pathways in the indicated CRISPR cell lines. Levels of pIGF-1R, total IGF-1R, pAKT, total AKT, pMEK, total MEK, pERK, and total ERK shown after 4 days ± 0.5 μM BRAFi or 0.5 μM BRAFi + 0.5 μM IGF-1R/IRi. GAPDH used as a loading control. **c** Indicated cell lines were seeded at the same density and cultured for 1 week in presence of the indicated compounds (top). Relative cell proliferation (percentage) of the same cells, $n = 3$. Data are mean ± SEM; ****$P < 0.0001$; Paired two-tailed Student's $t$-test was performed for comparisons (bottom). **d** L-C-B and SIRT6.2-7 cells were treated with DMSO, IGF-1R/IRi (2 μM), BRAFi (0.5 μM), or both at 0.5 μM for 72 h. Percentages of Annexin-V positive cells indicated, $n = 3$. Data are mean ± SEM; *$P < 0.05$, **$P < 0.01$ Paired two-tailed Student's $t$-test was performed for comparisons. **e** Quantification of tumor volume in nude mice bearing xenograft tumors of SIRT6.2–7 cells and that were fed a control, vemurafenib (PLX)-containing diet (100 mg/Kg), linsitinib (50 mg/Kg for 2 days, then 25 mg/Kg), or vemurafenib (PLX) + linsitinib (100 mg/Kg + 50 mg/Kg for 2 days, then 100 mg/Kg + 25 mg/Kg). Mann–Whitney test was performed for comparison between the three groups of mice treated. Data are mean ± SEM. *$P < 0.05$, **$P < 0.01$. $n = 5$-7

Supplementary Fig. 8c, d). We confirmed these results in additional SIRT6-edited clones (Supplementary Fig. 8c, d). To evaluate the clinical implications of our findings, we performed xenograft experiments with SIRT6.2–7 cells. While BRAFi stabilized growth in xenografted tumors, the combination treatment of BRAFi + linsitinib caused significant tumor regression (Fig. 4e). Together, these data indicate that SIRT6 haploinsufficiency allows BRAF[V600E] melanoma cells to survive in the presence of MAPKi by increasing IGFBP2 expression, which in turn activates IGF-1R/IR and downstream AKT signaling, that can be blocked by IGF-1R/IR inhibition.

**IGFBP2 is a potential biomarker for MAPKi resistance in melanoma**. By probing The Cancer Genome Atlas (TCGA), we found significant upregulation of IGFBP2 mRNA and protein in melanoma harboring BRAF mutation compared to other mutational subtypes (Fig. 5a). To confirm these findings, we performed IGFBP2 immunohistochemistry (IHC) on treatment-naive tumor biopsies from 34 melanoma patients (21 BRAF[V600E/K] and 13 non-BRAF mutant tissues) (Supplementary Data 5). Strikingly, we found significantly increased IGFBP2 in BRAF[V600E/K] tumors

compared to non-BRAF mutant tumors (Fig. 5b). Next, we performed IGFBP2 IHC on matched tumor biopsies from five patients with BRAF[V600E/K] metastatic melanoma treated with dabrafenib + trametinib taken before treatment (Pre), early during treatment (EDT), and at disease progression (Prog)[47] (Supplementary Data 5). Four of the five patients developed resistance to MAPKi during the course of therapy. Biopsies from Patients 1 and 2 were low/negative for IGFBP2 (Supplementary Fig. 9a). Patient 1 showed a complete response, while Patient 2 had stable disease with an acquired MEK1 mutation[48]. Strikingly, Patients 3, 4, and 5 who succumbed to their disease, showed high levels of IGFBP2 at baseline, which decreased at EDT biopsy, but returned upon progression (Fig. 5c, Supplementary Data 5). This is consistent with our findings that point towards a role for IGFBP2 in MAPKi resistance.

To evaluate whether SIRT6 and IGFBP2 levels anti-correlate in human melanoma tissues, we performed SIRT6 IHC on 21 BRAF[V600E/K] melanomas stained for IGFBP2, and observed a statistically significant inverse correlation (Spearman's $r = -0.462$; $P = 0.035$; Fig. 5d, e). Moreover, analysis of TCGA data for primary melanomas revealed that 20% display high SIRT6

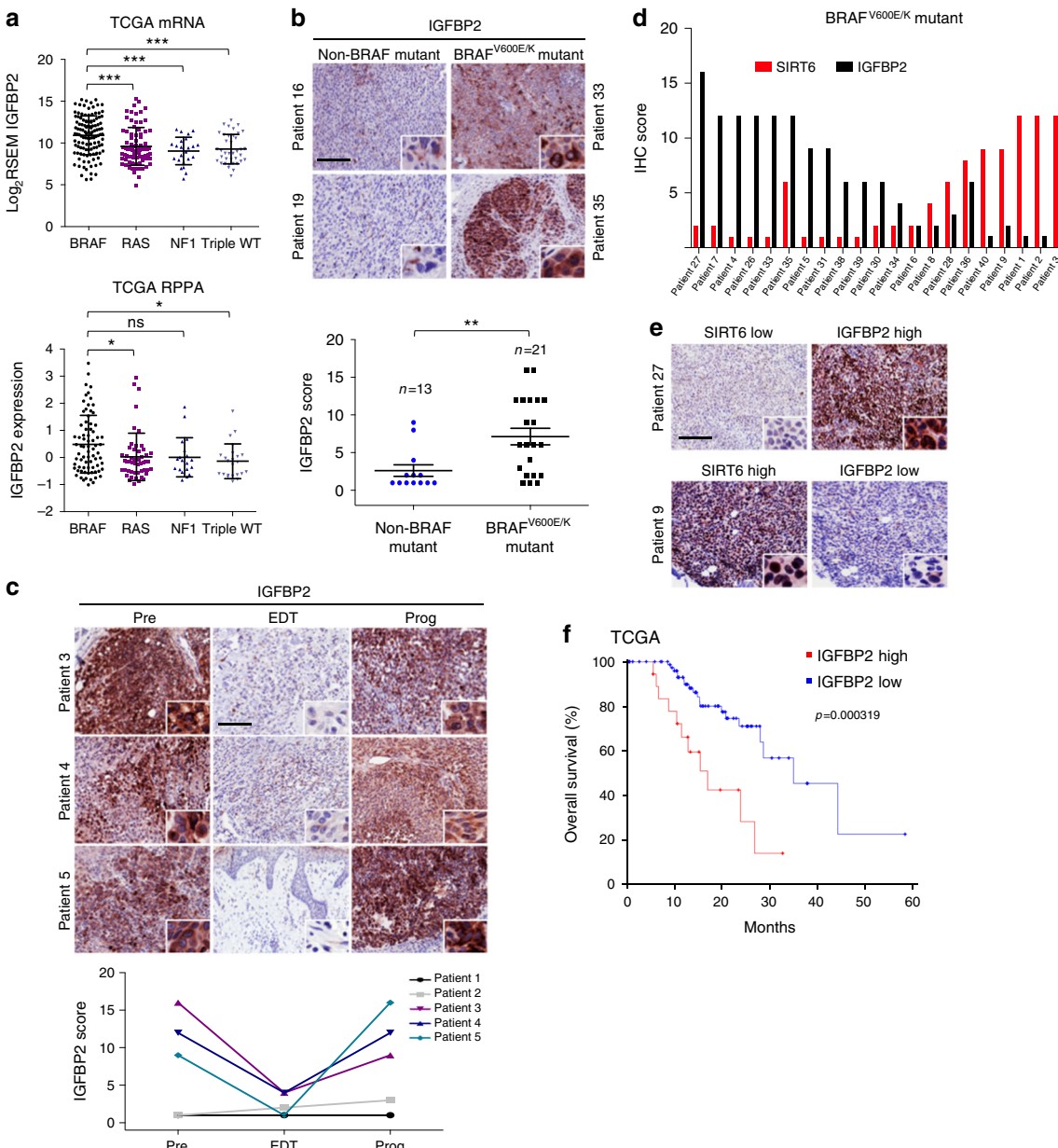

**Fig. 5** IGFBP2 represents a potential biomarker for MAPKi resistance in BRAF mutant melanoma. **a** IGFBP2 mRNA expression from TCGA samples classified by mutational status (top). IGFBP2 protein expression (RPPA) from TCGA samples classified by mutational status (bottom). Data are mean ± SD; *$P < 0.05$, ***$P < 0.001$; One-way ANOVA was used for comparisons. **b** IHC (immunohistochemistry) for IGFBP2 in non-BRAF mutant or BRAF$^{V600E/K}$ mutant melanoma tissue (Supplementary Data 5). Images ×10 magnification; insets ×40 magnification. Scale bar represents 100 μm. Scores obtained by multiplying the percentage of positively stained cells (1–4) by intensity of stain (1–4). Data are mean ± SEM; **$P < 0.01$; Mann–Whitney (two-tailed) was performed for comparison. **c** IHC for IGFBP2 in matched tumor samples (Patients 3, 4, and 5; Supplementary Data 5) taken before treatment (Pre), early during treatment (EDT), and on disease progression (Prog). Images ×10 magnification; insets ×40 magnification. Scale bar represents 100 μm. Scores obtained as in **b**. **d** IHC scores of SIRT6 and IGFBP2 in 21 BRAF$^{V600E/K}$ melanoma tissues (Supplementary Data 5); Scores obtained as in **b**; Spearman's $r = -0.462$; $P = 0.035$. **e** Representative IHC for SIRT6 and IGFBP2 in BRAF$^{V600E/K}$ melanoma tissues. Images ×10 magnification; insets ×40 magnification. Scale bar represents 100 μm. Scores obtained as in **b**. **f** Overall survival stratified by IGFBP2 protein expression (RPPA) from TCGA samples (primary tumors, $n = 92$). High IGFBP2 protein level in red; $z$-score threshold > 0.5 or low IGFBP2 protein level in blue; $z$-score threshold < 0.5. $P$-values, log rank test

mRNA, which anti-correlates with IGFBP2 (via RPPA) (Supplementary Fig. 9b), suggesting a repressive role of SIRT6 on IGFBP2 expression in melanoma. Finally, we found that IGFBP2 protein and mRNA levels are associated with poor prognosis in primary melanoma (Fig. 5f, Supplementary Fig. 10a, b), supporting IGFBP2 as a potential biomarker for MAPKi resistance in melanoma.

## Discussion

Mechanisms of MAPKi resistance in BRAF$^{V600}$-mutant melanoma have mainly focused on components of the ERK signaling pathway, however, little is known about the epigenetic regulators involved in this process. We hereby present a chromatin-focused CRISPR–Cas9 screen to identify factors that play a critical role in BRAF$^{V600}$-mutant melanoma resistance to MAPKi, and

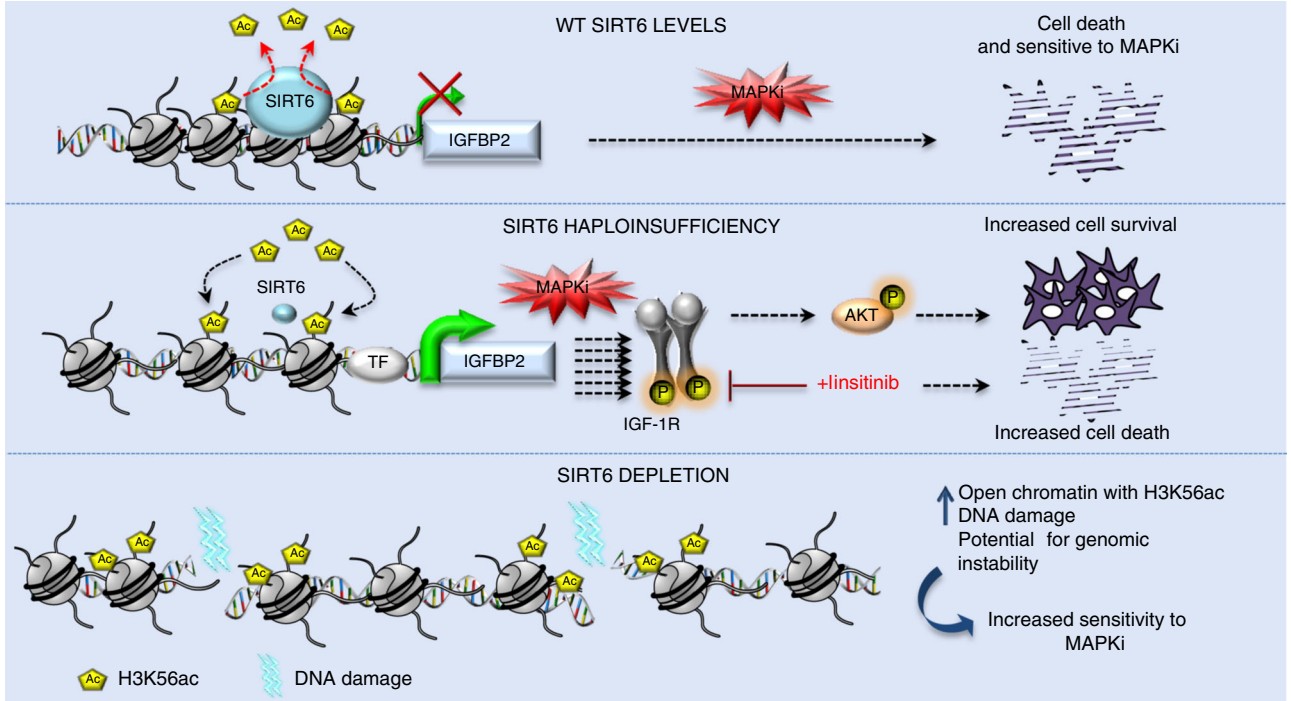

**Fig. 6** Model: SIRT6 levels regulate resistance or sensitivity to MAPKi. (Top) Under baseline conditions, SIRT6 deacetylates H3K56ac at the IGFBP2 locus to maintain low expression levels and low IGF-1R/AKT pathway signaling, impairing cell survival in the presence of MAPKi. (Middle) In the context of SIRT6 haploinsufficiency, IGFBP2 expression is enhanced via increased chromatin accessibility and H3K56 acetylation at the *IGFBP2* locus. Activation of the IGF-1 receptor (IGF-1R) and downstream AKT signaling allows melanoma cells to persist in the presence of MAPKi. (Bottom) SIRT6 depletion triggers significant chromatin reorganization reflected by increased open chromatin and H3K56ac at these sites, coupled to DNA damage accumulation. This potential for genomic instability increases the sensitivity to MAPKi

identified enzymes that mediate histone acetylation. We found that SIRT6 haploinsufficiency in BRAF$^{V600E}$ melanoma cells decreases sensitivity to MAPKi, independent of the ERK signaling pathway. SIRT6 haploinsufficiency allows cells to survive via IGFBP2 expression, which in turn activates IGF-1R and downstream AKT signaling in the presence of MAPKi (see model, Fig. 6).

In mammalian cells, the SIRTUIN family is composed of seven members (SIRT1-7), having distinct subcellular localization and functions in key biological processes[49,50]. SIRT1, 2, and 6 are considered the nuclear SIRTUINs and are chromatin-bound[50]. Interestingly, an shRNA screen identified that SIRT2 depletion conferred resistance to vemurafenib in BRAF$^{V600E}$ melanoma cells. In contrast to SIRT6, however, SIRT2 loss reactivated ERK signaling[51]. Recently, a genome-wide CRISPR–Cas9 screen uncovered enrichment of sgRNAs targeting members of the STAGA HAT complex in the presence of vemurafenib[25], consistent with a critical role for histone acetylation in melanoma drug resistance.

The link between SIRTUINs and IGF signaling has been reported in multiple systems. For example, SIRT6 acts at the chromatin level to regulate the transcriptional status of insulin/IGF-AKT signaling-related genes to prevent cardiac hypertrophy and heart failure[52]. Previous studies have also implicated increased activation of the AKT signaling pathway in promoting MAPKi resistance[53–55]. Interestingly, Villanueva et al., reported IGF-1R/PI3K signaling to play a role in MAPKi-resistant melanoma and hypothesized that additional factors (e.g., IGFBPs) were essential to engage this process. Here, through an unbiased screening approach, we identified SIRT6-mediated transcriptional regulation of IGFBP2, and consequent activation of IGF-1R/AKT, to play a role in melanoma MAPKi resistance. Therefore, we

provide a deeper mechanistic understanding of the upstream activators of IGF1R/AKT signaling in this process.

Our study suggests that SIRT6 haploinsufficiency enhances IGF-1R/AKT signaling via transcriptional control of *IGFBP2* in the presence of MAPKi (Fig. 6). Nevertheless, we cannot exclude that other genes control this signaling pathway. For example, we also uncovered *AEBP1* expression to be upregulated. *AEBP1* was previously implicated in resistance to BRAFi in melanoma, however, its expression is driven by hyperactivation of the PI3K/AKT pathway[42]. Additionally, no SIRT6-binding sites were identified around the *AEBP1* TSS in our study. Thus, the increase of *AEBP1* expression we observed is most likely a consequence of the activation of the IGF-1R/AKT survival pathway, rather than via direct regulation by SIRT6.

In contrast to SIRT6 haploinsufficient melanoma cells, melanoma cells lacking SIRT6 undergo chromatin reorganization reflected by increased open chromatin and H3K56ac at these nucleosome-depleted sites (Supplementary Fig. 11a, b, Supplementary Data 4). Such potential for genomic instability is consistent with the increased DNA damage we detected (Fig. 6, Supplementary Fig. 3c), as well as impaired xenografted tumor growth (Supplementary Fig. 6c). Therefore, we could consider complete SIRT6 depletion (e.g., CRISPR) as a potential novel strategy to enhance melanoma cell sensitivity to current targeted MAPKi therapies.

Finally, a recent study identified *IGFBP2* as part of a gene signature in response to MAPKi in a population of cells referred to as drug-tolerant persisters[28]. Importantly, the burden of acquired melanoma resistance emerges from such tumor subpopulations. Thus, early treatment to eradicate this population is key to delay or prevent drug resistance. While IGFBP2 is overexpressed in various tumors, including melanoma[44,45,56,57], and is

part of a gene signature of MAPKi drug-tolerant persisters[28], there is little evidence for its use as a biomarker[58]. We observed that IGFBP2 protein levels correlated with resistance to MAPKi in several BRAF$^{V600}$-mutant melanoma cell lines, and is associated with poor prognosis in primary melanomas (Supplementary Fig. 8b, Fig. 5f and Supplementary Fig. 10a, b). Interestingly, IGFBP2 has been implicated as a candidate diagnostic for heart failure with a high sensitivity and specificity by urine proteomic analyses[59], clearly highlighting its use as a potential biomarker for melanoma MAPKi resistance. In sum, our data strongly suggest that co-targeting of MAPK and IGF-1R pathways can prevent/delay resistance to targeted MAPKi therapies, particularly for patients with high levels of IGFBP2.

## Methods

**Generation of sgRNA library**. We used guide sequences provided by George Church lab[60] (Bioinformatically computed genome-wide resource of candidate unique gRNA targets in human exons is available here: http://arep.med.harvard.edu/human_crispr/). The custom oligonucleotide library was reconstituted in water to a final concentration of 0.01 pmol/μL and PCR-amplified using Q5 Hot Start Polymerase (New England Biolabs). The PCR-amplified library was then purified (Qiagen), digested with BbsI (New England Biolabs) at 37 °C overnight, and purified by electrophoresis on a 2% agarose gel and recovered using gel extraction kit (Qiagen). The library oligonucleotides were then cloned downstream of the human U6 promoter in a lentiviral vector containing EGFP downstream of the human PGK promoter (pLKO.1-EGFP). The vector backbone was digested with AgeI and EcoRI, treated with FastAP Thermosensitive Alkaline Phosphatase (Thermo Scientific), purified on a 1% agarose gel by gel extraction (Qiagen). Ligation was performed using Quick Ligase kit (New England Biolabs). To ensure library diversity, colonies were collected from 15 bacterial plates after transformation of 10-beta electrocompetent cells (New England Biolabs). The pool of plasmids was prepared for infection using an endotoxin-free Maxi prep kit (Qiagen). The library targeting chromatin-related factors contains (see Supplementary Data 1 for details).

**CRISPR–Cas9 screen**. SKMel-239 were first infected with the lentiCas9-Blast (L-C-B) (Addgene #52962) and selected with blasticidin (10 μg/mL). Lentiviral vectors were produced as previously described[61,62]. Viral titer was estimated on 293T cells by limiting dilution. Cells were then infected with the sgRNA library at a low MOI (<1) to ensure a single sgRNA vector per cell. After 4 days of infection, cells were analyzed by flow cytometry and <20% of cells were EGFP-positive, corresponding to single vector copy. EGFP-positive cells were expanded for 10 days, plated at low density and cultured in the presence of DMSO, 2 μM of dabrafenib or 100 nM + 1 nM of dabrafenib + trametinib (3 or 4 plates per condition) for 42 days. A fraction of cells were collected at day 0 to ensure a proper coverage of sgRNAs. Medium was changed every 3 days. At day 42, cells from all conditions were collected and genomic DNA was extracted. Since melanin pigment may interfere with DNA-and/or RNA-based molecular profiling[63], we purified the samples using the OneStep$^{TM}$ PCR inhibitor Removal Kit (Zymo Research). The integrated sgRNAs were then amplified by PCR with primers containing multiplexing barcodes and adaptors and sequenced on the Illumina NextSeq500. Hits were selected based on the log2 fold change of sgRNA reads at day 42 in presence of the indicated drug(s) vs. DMSO-treated cells at day 42, and their presence in both screens. Analyses and plots of the sequencing data were conducted using Prism 6 software (GraphPad Software) and Rank Products Analysis to determine P values.

**Cell culture**. SKMel-239 (MSKCC) cells were cultured in RPMI; 501Mel (Yale University), Mel888 (Stuart Aaronson), SKMel-147 (MSKCC), and SKMel-28 (ATCC) were cultured in DMEM. Human melanoma STCs (CM150 (patient 1), CM145 (patient 2), and CM143 (patient 3)) isolated pre- and post-treatment with vemurafenib were cultured as described[38]. 293T cells used for virus production were maintained in DMEM. All medium were supplemented with 10% FBS, 100 IU of penicillin and 100 μg/mL of streptomycin. Media for melanoma resistant cells contained 1 μM of vemurafenib (STCs) or as indicated. Dabrafenib and dabrafenib + trametinib resistant SKMel-239 cell lines were generated by seeding cells at low density and continuously exposed to 2 μM of dabrafenib or 100 nM of dabrafenib + 1 nM of trametinib for the combination. After ~6 weeks, resistant cell clones were derived and maintained in dabrafenib or on dabrafenib + trametinib.

**Plasmids and infections**. pLKO.1 vectors encoding shRNAs were purchased from Sigma (shSIRT6#1: TRCN0000050475, shSIRT6#2: TRCN0000050476, and shSIRT6#3: TRCN0000050477). The lentiCas9-Blast plasmid (#52962) was purchased from Addgene to generate an SKMel-239 cell line stably expressing Cas9 used for the CRISPR screen. Infections were performed using standard procedures[8]. The sgRNAs of interest were cloned in a lentiviral vector containing EGFP downstream of the human PGK promoter (pLKO.1-EGFP) used for the library

generation. HA-fused complementary DNA (cDNAs) encoding IGFBP2 was cloned into the lentiviral vector VIRSP (gift of Aaronson lab).

**Whole-cell protein extractions, chromatin fractionation, and immunoblotting**. Cells were washed with PBS and lysed on ice for 5 min in NP40 buffer (50 mM Tris pH 7.5, 1% NP40, 150 mM NaCl, 10% Glycerol, 1 mM EDTA) supplemented with protease and phosphatase inhibitors (Roche). Lysates were centrifuged at 15,000 rpm for 15 min and the protein concentration was quantified using BCA (Pierce). Chromatin fractionation performed as described[8]. All lysates were freshly prepared and supplemented with Laemmli loading buffer with subsequent boiling for immunoblotting. The primary antibodies used for immunoblotting were anti-SIRT6 (1:1000, CST, #12486), anti-IGFBP2 (1:1000, Abcam, ab109284), anti-H3K4me3 (1:2500, Abcam, ab1012), anti-H3K9ac (1:1000, Abcam, Ab10812), anti-H3K27ac (1:1000, Abcam, ab4729), anti-H3K56ac (1:1000, Abcam, ab76307), anti-p-IGF-1R (1:200, Cell Signaling, #3918), anti-IGF-1R (1:1000, Cell Signaling, #3027), anti-p-AKT (1:1000, Cell Signaling, #4060), anti-AKT (1:1000, Cell Signaling, #9272), anti-p-MEK (1:1000, Cell Signaling, #9121), anti-MEK (1:1000, Cell Signaling, #2352), anti-p-ERK (1:1000, Cell Signaling, #4370), anti-ERK (1:1000, Cell Signaling, #9107), anti-KAT1 (1:1000, Santa-Cruz, sc-390562), anti-KAT2B (1:1000, Santa-Cruz, sc-13124), anti-GAPDH (1:1000, Santa-Cruz, sc-32233), anti-b-Actin (1:1000, Sigma, A5441), and anti-Vinculin (1:1000, Sigma, V9131). Uncropped western blots can be found in Supplementary Fig. 12.

**Compounds**. Dabrafenib (#S2807), Vemurafenib (#S1267), Trametinib (#S2673), and Linsitinib (OSI-906) (#S1091) were purchased from Selleck Chemicals.

**Cell proliferation, apoptosis, and DNA damage assays**. For short-term assays, cells were seeded in 24-well plates ($2 \times 10^4$ cells per well), allowed to adhere overnight and then incubated with media containing dabrafenib or dabrafenib + trametinib. After 72 h, the number of cells or the percentage of confluency were determined using the Countess II FL from Life technologies or the IncuCyte ZOOM from Essen BioScience respectively. For long-term assays, cells were seeded in 6-well plates at low density, allowed to adhere overnight, and cultured in the absence and/or presence of drugs as indicated. For Annexin V analysis, cells were stained with annexin-APC (BD Biosciences) and propidium iodide. Samples were subsequently analyzed on LSR Fortessa apparatus. The alkaline comet assay was performed as described with modifications[64]. Cells were treated with DMSO, 2 μM of BRAFi or 100 nM BRAFi + 1 nM MEKi for 4 days. $1 \times 10^5$ cells were trypsinized and diluted in 100 μL of 0.5% low melting point agarose and placed over pre-coated agarose slide. Cells were then lysed (2.5 M NaCl, 100 mM EDTA, 10 mM Tris, pH 10, 1% Triton, and 10% DMSO) for 24 h at 4 °C, incubated in electrophoresis buffer (300 mM NaOH, pH 13, 1 mM EDTA) for 30 min and subjected to electrophoresis in the dark for 25 min at 25 V. Slides were neutralized three times with Tris buffer (0.4 M Tris, pH 7.5) for 5 min each, dried with 100% ethanol and stained with ethidium bromide (20 μg/mL). Cells were analyzed in Nikon Eclipse ® microscope and ≥100 random cells per slide were analyzed using CellProfiler software. Cell images were segmented using a pixel intensity of 0.5 as threshold to generate masks matching the nucleoid. The comet tail was calculated subtracting the nucleoid-integrated intensity from the comet-integrated intensity. For each sample, a positive control with cells treated with Hydrogen peroxide ($H_2O_2$) (100 μM for 30 min at 25 °C) was analyzed concurrently. Experimental analysis was performed in a blinded fashion.

**Xenograft model**. All studies and procedures involving mice were performed following Massachusetts General Hospital IACUC guidelines. Six-week-old female athymic mice (NCr$^{nu/nu}$) were purchased from Taconic farms. Animals were allowed for a 1 week adaptation period upon arrival. For the vemurafenib sensitivity experiment, L-C-B, S6.2–7, and S4.1-1 cells ($2 \times 10^6$ in 0.2 mL of basal culture medium) were injected subcutaneously in the right lateral flank. Tumor dimensions were measured with calipers and volumes were calculated using the following formula:$(D \times d2)/2$, in which $D$ represents the large diameter of the tumor, and $d$ represents the small diameter of the tumor. When tumor volumes reached 80–120 mm$^3$, animals were randomly assigned to two groups, which were administered vemurafenib diet or control diet by the Research Randomizer at http://www.randomizer.org. Vemurafenib diet (5.67 g/kg body weight to achieve a 100 mg/kg body weight daily dose) and control diet were prepared at Harlan Laboratories (Madison, WI). For the vemurafenib and linsitinib combination experiment, S6.2–7 cells ($2 \times 10^6$ in 0.2 mL of basal culture medium) were injected subcutaneously in the right lateral flank. When tumor volume reached 80–120 mm$^3$, animals was randomly assigned to four groups, which were administered control diet, vemurafenib diet, control diet plus linsitinib, and vemurafenib diet plus linsitinib. Linsitinib (purchased from Selleck Chemicals) was prepared as 10 mg/mL in a 25 mM tartaric acid solution and administered (50 mg/kg for the first 2 days, and 25 mg/kg for the rest) by oral gavage once daily. Animals were monitored over a 14-day period and were killed when the tumor size reached 1.5 cm in any dimension. For the tumor growth experiment in Supplementary Fig. 6c, S6.2–7 and S6.1-1 cells ($2 \times 10^6$ in 0.2 mL of basal culture medium) were injected subcutaneously in the right lateral flank. Tumor dimensions were measured for 27 days.

**RNA extraction, RNA-seq, and qRT-PCR**. Total RNA was extracted using RNeasy Mini Kit (Qiagen). RNA for sequencing was processed into Poly A libraries (Illumina). Reverse transcription was performed with First-strand cDNA Synthesis (OriGene). qPCR reactions were performed in triplicate on CFX384 Touch™ (BioRad) using FastStart Universal SYBR Green Master (Roche). cDNA expression was normalized to GAPDH or B-actin levels. Each qPCR was performed on three independent biological replicates. Primer sequences can be found in Supplementary Data 6.

**RNA-sequencing analysis**. Libraries were sequenced on Illumina HiSeq2500 (~50 M reads, 100nt single end). Reads were aligned to the GRCh37/hg19 using STAR (version v2.4.1c)[65]. Transcriptome assemblies and differential expression ratios were performed using featureCount: subread 1.4.6-p2 and voom-limma: 3.26.9[66,67]. Genes were selected as following in all conditions: log2 Fold Change $\leq -1$ or $\geq 1$; and nominal $P$-value of $P$ value $\leq 0.05$. Box plots and Volcano plots were generated using Prism 6 software (GraphPad Software).

**GO analysis**. GO terms were obtained using Enrichr[68]. Combined score of top categories is shown for all plots.

**ChIP-sequencing analysis**. Chromatin for native ChIP of histone modifications was digested with Micrococcal nuclease (MNase) and immunoprecipitated as described[69]. For SIRT6, $40 \times 10^6$ cells (L-C-B and SIRT6.2–7) were crosslinked with 0.4% PFA for 10 min at RT. ChIP was performed as described[70]. Immunoprecipitations were performed with 10 µg of specific antibodies anti-SIRT6 (Abcam, Ab62739), anti-H3K4me1 (Abcam, ab8895), anti-H3K4me3 (Abcam, ab1012), anti-H3K9ac (Abcam, Ab10812), anti-H3K27ac (Abcam, ab4729), and anti-H3K56ac (Abcam, ab76307). Sequencing libraries were generated and barcoded for multiplexing as described[69]. Libraries were sequenced on NextSeq500 (50–75 bp single-end reads). Reads were trimmed for Illumina adapter sequences using in-house scripts and aligned to the GRCh37/hg19 using Bowtie (version 0.12.7) with parameters -l 65 -n 2 -S -best -k 1 -m 20. SIRT6 significant peaks were identified using MACS2 (version 2.1.1.2) and matching Input control was used to call peaks. Coverage tracks were generated from BAM files using deepTools (version 2.4.1) bamCoverage with parameters -normalizeUsingRPKM -binsize 10. Differential ChIP-seq-binding profiles for the histone marks H3K9ac and H3K56ac between L-C-B and SIRT6.2–7 cells were found by using MACS2 bdgdiff (parameters: -l 100 –g 100 or –l 250 –g 250 for H3K9ac and H3K56ac, respectively).

**ATAC sequencing**. All ATAC-seq libraries were prepared essentially as previously described with modifications[71]. Briefly, 50 K cells were resuspended in 1 mL of cold ATAC-seq resuspension buffer (RSB; 10 mM Tris-HCl pH 7.4, 10 mM NaCl, and 3 mM MgCl$_2$ in water). After centrifugation, cell pellets were resuspended in 50 µL of ATAC-seq RSB containing 0.1% NP40, 0.1% Tween-20, and 0.01% digitonin. This cell lysis reaction was incubated on ice for 3 min. Nuclei were isolated using 1 mL of ATAC-seq RSB containing 0.1% Tween-20 (without NP40 or digitonin) and centrifugation. Nuclei were resuspended in 50 µL of transposition mix (25 µL 2 × TD buffer[71], 2.5 µL transposase[72] (100 nM final), 16.5 µL PBS, 0.5 µL 1% digitonin, 0.5 µL 10% Tween-20, and 5 µL water). Transposition reactions were incubated at 37 °C for 30 min in a thermomixer with shaking. Reactions were cleaned up with Qiagen miniElute PCR Purification columns. ATAC-seq libraries were PCR-amplified with 5 to 7 cycles, size selected (100–800 bp), and purified using XPure magnetic beads[17]. Libraries were sequenced on NextSeq500 (75 bp paired-end reads).

**ATAC sequencing analysis**. Reads were aligned to the GRCh37/hg19 using Bowtie2 (version 2.1.0) with parameters –X 2000[73]. Bam files were processed (samtools version 1.6) by removing reads that: (1) aligned to the mitochondrial genome, (2) did not aligned to the nuclear genome, (3) with quality value $Q < 30$, and (4) were PCR duplicates (picard-tools-1.107). Significant peaks were called on merged bam files from all samples, using MACS2 call peaks (version 2.1.1.2) with parameters –nomodel –nolambda –keepdup all –slocal 10000. Peaks intersecting blacklisted regions were removed. Top 100 K peaks based on –LOG10qValue (MACS2) were considered for downstream analyses. Data was normalized by the total reads in peaks in TSS of coding genes (Gencode V19) (bedtools v2.17.0). Normalized coverage tracks were generated using deepTools bamCoverage (version 2.4.1) with parameters –scaleFactor (as determined by normalization) –skipNon-CoveredRegions –binsize 10. Peaks with FC > 2 and FC < 0.5 in read counts were considered differential[74].

**Reverse phase protein array**. Cellular proteins were denatured by 1% SDS (with Beta-mercaptoethanol) and diluted in five twofold serial dilutions in dilution lysis buffer. Serial diluted lysates were arrayed on nitrocellulose-coated slides (Grace Bio Lab) by Aushon 2470 Arrayer (Aushon BioSystems). Total 5808 array spots were arranged on each slide including the spots corresponding to serial diluted: (1) Standard Lysates; (2) positive and negative controls prepared from mixed cell lysates or dilution buffer, respectively. Each slide was probed with a validated primary antibody plus a biotin-conjugated secondary antibody. Only antibodies with a Pearson correlation coefficient between RPPA and western blotting of >0.7 were used for RPPA. Antibodies with a single or dominant band on western blotting were further assessed by direct comparison to RPPA using cell lines with differential protein expression or modulated with ligands/inhibitors or siRNA for phospho- or structural proteins, respectively. The signal obtained was amplified using a Dako Cytomation–Catalyzed system (Dako) and visualized by DAB colorimetric reaction. The slides were scanned, analyzed, and quantified using a customized-software to generate spot intensity. Each dilution curve was fitted with a logistic model; Supercurve Fitting developed by the Department of Bioinformatics and Computational Biology in MD Anderson Cancer Center, http://bioinformatics.mdanderson.org/OOMPA. This fits a single curve using all the samples (i.e., dilution series) on a slide with the signal intensity as the response variable and the dilution steps are independent variable. The fitted curve is plotted with the signal intensities—both observed and fitted—on the y-axis and the log2-concentration of proteins on the x-axis for diagnostic purposes. The protein concentrations of each set of slides were then normalized for protein loading. Correction factor was calculated by: (1) median-centering across samples of all antibody experiments; and (2) median-centering across antibodies for each sample. The heatmap was generated using Cluster 3.0 and visualized in Treeview.

**Patient samples and IHC**. Formalin-fixed paraffin-embedded (FFPE) tissue sections of non-BRAF or BRAF$^{V600E/K}$ mutant primary tumors and metastatic resections were obtained from Mount Sinai Hospital Department of Pathology, Dermatopathology Division. The Institutional Review Board at the Icahn School of Medicine at Mount Sinai approved this study (project number HSD08-00565). Formalin-fixed paraffin-embedded (FFPE) tissue sections collected to evaluate IGFBP2, as well as clinical information from patients 1–5 were obtained under the auspices of the Treat Excise Analyze for Melanoma (TEAM) study at the Melanoma Institute Australia (Royal Prince Alfred Hospital Research Ethics Committee Protocol No. X15-0418/X10-0305 and HREC/10/RPAH/539). Written consent was obtained from all patients under approved Human Research ethics committee protocols (above). Slides containing FFPE tissue sections were manually deparaffinized through xylene and graded ethanol washes. After heat-induced epitope retrieval with citrate buffer (pH 6.0), the samples were incubated with anti-IGFBP2 antibody (1:100, Abcam, Ab109284) or anti-SIRT6 antibody (1:200, Cell Signaling, #12486) overnight followed by incubation with a universal secondary antibody (ImmPRESS™ HRP REAGENT KIT Vector #MP-7500) for 20 min. Detection was performed using nova red for 8 min and the slides were counterstained with hematoxylin. All tissues were analyzed in a blinded fashion by two pathologists (R.S. and M.S.G.).

**Data availability**. The raw and processed sequencing data reported in this paper has been deposited in the GEO with the following accession number: GSE102813 (https://www.ncbi.nlm.nih.gov/geo/query/acc.cgi?acc = GSE102813). All other remaining data are available within the Article and Supplementary Files, or available from the authors upon request.

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

## Acknowledgements

We thank Chiara Vardabasso, Asif Chowdhury, Dan Filipescu, Andriana Kotini, Philip Friedlander, Kaitlyn Fragogiannis, and Steven Chen for advice and assistance. We acknowledge Richard Kefford and other colleagues at Melanoma Institute Australia, the Department of Tissue Pathology and Diagnostic Oncology, Royal Prince Alfred Hospital, Hospital and Crown Princess Mary Cancer Centre Westmead Hospital, Sydney, Australia. Funding support was provided by the Australian National Health and Medical Research Council, NHMRC 633004, Australian National Health, and Medical Research Council Fellowships to G.V.L. and R.A.S., NIH grant U54HL127624 to A.M., NIH R01-CA166717 to B.Z., R01AI104848, and R33CA182377 to B.B., Scientific Computing at ISMMS, Office of Research Infrastructure of the NIH to ISMMS (S10OD018522), Tisch Cancer Institute P30 CA196521, La Roche-Posay North American Foundation, American Skin Association, and The Philippe Foundation Inc. to T.S., and the Pershing Square Sohn Cancer Research Alliance and Department of Defense (W81XWH-14-1-0230) to E.B.

## Author contributions

T.S. and E.B. conceived this study. T.S. performed CRISPR–Cas9 screen with the support of A.W. and B.B. A.W. generated sgRNA libraries and virus. T.S. performed cell based assays with S.C. and F.G.G., immunoblots, IHC, RNA-seq, and ChIP-seq studies. D.H. assisted with ChIP-seq and performed ATAC-seq, Z.W. performed RNA-seq analysis with the support of A.M., F.G.G. performed comet assays, M.L. performed xenografts with support of B.Z., S.G., and P.H. provided STCs, G.V.L. and R.A.S. provided pre-, EDT, and progression melanoma tissues, R.S. and M.S.G. acquired tissues and scored all IHC. T.S. and E.B. wrote this manuscript with feedback from all other authors.

## Additional information

**Competing interests:** The authors declare no competing interests.

