## [Peer Review File · Nature Communications]

Reviewers' comments:

Reviewer #1 (Remarks to the Author):

The authors have done a good job of responding to the various criticisms and suggestions.

Reviewer #2 (Remarks to the Author):

In this revised version of their manuscript, Bernstein and colleagues did major efforts in responding to the previous reviewers' concerns, and the manuscript is significantly improved. Particularly, the new data on the KO cells (indicating that they behave different from the haploinsufficient cells because they exhibit increased DNA damage), and the additional data on patients, strongly support their model. The paper is highly significant in that it supports a putative novel combinatorial treatment for melanoma particularly in the context of BRAF-mutant tumors. I have no more concerns.

Reviewer #3 (Remarks to the Author):

In the new version of the manuscript, the authors have answered most of my previous concerns satisfactorily. Their changes to the previous version have increased significantly the overall quality of the work, which should be relevant for a wide range of basic and translational researchers. The evidences reported by the authors strongly support a novel therapeutic approach for treatment of BRAFV600E melanomas through modulation of IGF signaling. I only have two remaining issues. The first one is that as I mentioned in the previous review, the authors have not include any direct evidence to demonstrate a direct role of IGFBP2 as a mediator of the MAPKi resistance induced by SIRT6 partial depletion on BRAFV600E melanomas. This is a very relevant issue for the mechanism proposed. I agree with the authors that the link between IGF signaling and SIRT6 has been well demonstrated, but there is no direct evidence about a direct role of IGFBP2 as a mediator or at least contributor of this SIRT6-associated resistance. The inverse correlation between SIRT6 and IGFBP2 in the IHCs, the effect of IGFBP2-/ +insulin in WT cells (Fig 4) and the clear effect of linsitinib are solid evidences, but are not demonstration of this direct implication. I would suggest to perform a similar experiment as in 4a but using SIRT6.2-7 cells. If the model is correct there should be a clear effect on this MAPKi resistance. The second issue is regarding SIRT6.1-1 cells. If I understand it well, the evidences seem to suggest that these cells are impaired because SIRT6 complete depletion also induces high levels of DNA damage and genome instability, and not because the resistance mechanism depends on a specific restricted dose of SIRT6. This seems a reasonable explanation to explain the specific effect of SIRT6 haploinsufficiency on the development of resistance to MAPKi. Since the haploinsufficiency effect is a central issue in the manuscript, for the sake of clarification I would suggest including the case of SIRT6 complete depletion (SIRT6.1-1) in the model of figure 4h.

Response to Reviews of Manuscript # NCOMMS-18-12679-T

June 14, 2018

Dear Reviewers,

We thank you for your thoughtful feedback of our manuscript "*SIRT6 haploinsufficiency induces BRAFV600E melanoma cell resistance to MAPK inhibitors via IGF signaling*". Previously, all reviewers were overall very positive regarding: (i) our approach to discover novel melanoma resistance genes, (ii) the significance of our findings for melanoma therapeutics and, (iii) our thoroughness, interpretation and presentation of the data.

Reviewers #1 and #2 are now **completely satisfied** with our revisions, and Reviewer #3 had some remaining points.

We have underlined the changes in the manuscript to facilitate identification of text changes as well as the new data. Below you will find a point-by-point response to the reviewers, our responses are highlighted in blue.

Reviewers' comments:

Reviewer #1 (Remarks to the Author):

The authors have done a good job of responding to the various criticisms and suggestions.

We thank Reviewer #1 for reviewing our manuscript and we are pleased to have completely addressed his/her comments.

Reviewer #2 (Remarks to the Author):

In this revised version of their manuscript, Bernstein and colleagues did major efforts in responding to the previous reviewers' concerns, and the manuscript is significantly improved. Particularly, the new data on the KO cells (indicating that they behave different from the haploinsufficient cells because they exhibit increased DNA damage), and the additional data on patients, strongly support their model. The paper is highly significant in that it supports a putative novel combinatorial treatment for melanoma particularly in the context of BRAF-mutant tumors. I have no more concerns.

We thank Reviewer #2 for reviewing our manuscript and for highlighting our efforts to improve to the quality of our study. We are delighted to have fully addressed his/her comments.

Reviewer #3 (Remarks to the Author):

In the new version of the manuscript, the authors have answered most of my previous concerns satisfactorily. Their changes to the previous version have increased significantly the overall quality of the work, which should be relevant for a wide range of basic and translational researchers. The evidences reported by the authors strongly support a novel therapeutic approach for treatment of BRAFV600E melanomas through modulation of IGF signaling. I only have two remaining issues.

The first one is that as I mentioned in the previous review, the authors have not include any direct evidence to demonstrate a direct role of IGFBP2 as a mediator of the MAPKi resistance induced by SIRT6 partial depletion on BRAFV600E melanomas. This is a very relevant issue for the mechanism proposed. I agree with the authors that the link between IGF signaling and SIRT6 has been well demonstrated, but there is no direct evidence about a direct role of IGFBP2 as a mediator or at least contributor of this SIRT6-associated resistance. The inverse correlation between SIRT6 and IGFBP2 in the IHCs, the effect of IGFBP2-/+insulin in WT cells (Fig 4) and the clear effect of linsitinib are solid evidences, but are not demonstration of this direct implication. I would suggest to perform a similar experiment as in 4a but using SIRT6.2-7 cells. If the model is correct there should be a clear effect on this MAPKi resistance.

The second issue is regarding SIRT6.1-1 cells. If I understand it well, the evidences seem to suggest that these cells are impaired because SIRT6 complete depletion also induces high levels of DNA damage and genome instability, and not because the resistance mechanism depends on a specific restricted dose of SIRT6. This seems a reasonable explanation to explain the specific effect of SIRT6 haploinsufficiency on the development of resistance to MAPKi. Since the haploinsufficiency effect is a central issue in the manuscript, for the sake of clarification I would suggest including the case of SIRT6 complete depletion (SIRT6.1-1) in the model of figure 4h.

We thank Reviewer #3 for reviewing our manuscript and for highlighting that we have significantly improved the quality of our study. However, we respectfully disagree with reviewer #3's additional concern about the role of IGFBP2 in mediating resistance. While the reviewer requests "direct evidence to demonstrate a direct role of IGFBP2 as a mediator of the MAPKi resistance induced by SIRT6 partial depletion on BRAF^{V600E} melanomas", the suggestion to perform the same experiment as in Fig. 4a using SIRT6.2-7 cells is puzzling. Because SIRT6 haploinsufficient cells (e.g. SIRT6.2-7) already display high levels of IGFBP2 and a decreased sensitivity to MAPKi compared to control cells, is the proposed experiment expected to reveal even more resistance?? It is possible that the reviewer meant to suggest SIRT6.1-1 cells (e.g. SIRT6 KO) to add back IGFBP2 + insulin, however since these cells display significant DNA damage, we don't expect IGFBP2 + insulin to promote resistance. Therefore, the proposed experiments are unlikely to yield informative results.

Please find additional points below that will hopefully clarify Reviewer #3's concerns:

-By overlapping independent and unbiased RNA-seq/proteomic/ChIP-seq data sets upon SIRT6 haploinsufficiency, IGFBP2 was consistently identified as a key player in MAPKi resistance (**Figure 3**).

-The IGFBP2 locus is a direct SIRT6 direct target (via ChIP-seq) that is upregulated in SIRT6 haploinsufficient cells (via RNA-seq), has more open chromatin in these cells (via ATAC-seq), and gains H3K56ac (ChIP-seq) (**Figure 3f**).

-IGFBP2 overexpression and insulin treatment in control cells decreases sensitivity to MAPKi (**Figure 4a**).

-Inhibition of the IGF-1R/IR signaling pathway (mediated by IGFBPs and IGFs) impedes the resistance of SIRT6 haploinsufficient cells both *in vitro* and *in vivo* (**Figure 4b-e**).

-Melanoma cell lines with varying levels of IGFBP2 positively correlate with resistance to MAPKi, i.e. higher IGFBP2 = increased resistance (**Supplementary Figure 8b**).

-IGFBP2 represents a useful biomarker for resistance as determined by IHC staining in matched patient samples pre-treatment, on-treatment, and upon resistance to BRAFi+MEKi therapy (**Figure 5c**). It is also associated with poor prognosis in melanoma patients (**Figure 5f**, **Supplementary Figure 10a-b**).

-We demonstrate a statistically significant inverse correlation of SIRT6 and IGFBP2 using human melanoma patients samples IHCs (**Figure 5d, e**) as well as in primary tumors samples from TCGA (**Supplementary Figure 9b**).

We hope the above comments fully address Reviewer #3's remaining concerns. As suggested, we have incorporated the SIRT6 KO phenotype into our model for clarification (new **Figure 6**), and added text to the Discussion section suggesting that other genes may be important downstream of SIRT6 (besides IGFBP2).

REVIEWERS' COMMENTS:

Reviewer #3 (Remarks to the Author):

The authors have now addressed all my issues satisfactorily. The experiment I proposed was meant to demonstrate that IGFBP2 by itself could increase further the resistance to MAPKi induced by SIRT6 haploinsufficiency. However, I understand the main handicap associated to the experiment, because as the authors suggest, to observe further resistance under these conditions could be technically very challenging. Considering these issues together with the amount of solid evidence already included in the manuscript and the relevance of the work, I am now happy to recommend publication in Nat communications.

Response to Reviews of Manuscript # NCOMMS-18-12679-T

July 5th, 2018

Reviewer #3 (Remarks to the Author):

The authors have now addressed all my issues satisfactorily. The experiment I proposed was meant to demonstrate that IGFBP2 by itself could increase further the resistance to MAPKi induced by SIRT6 haploinsufficiency. However, I understand the main handicap associated to the experiment, because as the authors suggest, to observe further resistance under these conditions could be technically very challenging. Considering these issues together with the amount of solid evidence already included in the manuscript and the relevance of the work, I am now happy to recommend publication in Nat communications.

We thank Reviewer #3 for reviewing our manuscript one last time, and we are pleased to have now completely addressed his/her comments.